# Impact of gallbladder hypoplasia on hilar hepatic ducts in biliary atresia
Nanae Miyazaki [1,10], Shohei Takami[1,2,10], Mami Uemura[1,3], Hironobu Oiki[1,2,4], Masataka Takahashi [5], Hiroshi Kawashima[4], Yutaka Kanamori[5], Takako Yoshioka[6], Mureo Kasahara [7], Atsuko Nakazawa[8], Mayumi Higashi[9], Ayaka Yanagida [1], Ryuji Hiramatsu[1], Masami Kanai-Azuma[3], Jun Fujishiro[2] & Yoshiakira Kanai [1] ✉

## Abstract

**Background** Biliary atresia (BA) is an intractable disease of unknown cause that develops in the neonatal period. It causes jaundice and liver damage due to the destruction of extrahepatic biliary tracts,. We have found that heterozygous knockout mice of the *SRY related HMG-box 17 (Sox17)* gene, a master regulator of stem/progenitor cells in the gallbladder wall, exhibit a condition like BA. However, the precise contribution of hypoplastic gallbladder wall to the pathogenesis of hepatobiliary disease in *Sox17* heterozygous embryos and human BA remains unclear.

**Methods** We employed cholangiography and histological analyses in the mouse BA model. Furthermore, we conducted a retrospective analysis of human BA.

**Results** We show that gallbladder wall hypoplasia causes abnormal multiple connections between the hilar hepatic bile ducts and the gallbladder-cystic duct in *Sox17* heterozygous embryos. These multiple hilar extrahepatic ducts fuse with the developing intrahepatic duct walls and pull them out of the liver parenchyma, resulting in abnormal intrahepatic duct network and severe cholestasis. In human BA with gallbladder wall hypoplasia (i.e., abnormally reduced expression of SOX17), we also identify a strong association between reduced gallbladder width (a morphometric parameter indicating gallbladder wall hypoplasia) and severe liver injury at the time of the Kasai surgery, like the *Sox17*-mutant mouse model.

**Conclusions** Together with the close correlation between gallbladder wall hypoplasia and liver damage in both mouse and human cases, these findings provide an insight into the critical role of SOX17-positive gallbladder walls in establishing functional bile duct networks in the hepatic hilus of neonates.

## Plain Language Summary

Biliary atresia (BA) is a disease in newborns that causes a serious liver condition due to damage to the bile ducts (the pathways that carry bile juice). Although reduced function of a key gene called *Sox17*, which is essential for forming the gallbladder wall, has been observed in some BA cases, the link between gallbladder issues and liver damage is unknown. This study has shown how damage spreads through the bile ducts in the liver around the time of birth when there are problems in the gallbladder wall due to reduced SOX17 function. The findings indicate that proper growth of the gallbladder wall during this critical period is essential for forming a normal network of bile ducts in the developing liver. This discovery is promising for early diagnosis and better treatment of BA in newborns.

In the developing hepatobiliary system, bile is produced and secreted by hepatocytes from the late organogenic stage and then passes through the canaliculi to the interconnected ducts of increasing diameter from the liver to the duodenum. The intrahepatic bile ducts originate from the liver bud at the ventral endoderm, which appears on the 26th day after fertilization in human[1] (embryonic days [E] 8.5 in mouse[2,3]). In contrast, the gallbladder bud at the caudal region of liver bud, which appears on the 29th day after fertilization in human[1] (E9.5 in mouse[2,3]), develops into the gallbladder and

[1]Department of Veterinary Anatomy, The University of Tokyo, Bunkyo-ku, Tokyo, Japan. [2]Department of Pediatric Surgery, the University of Tokyo, Bunkyo-ku, Tokyo, Japan. [3]Center for Experimental Animals, Tokyo Medical and Dental University, Bunkyo-ku, Tokyo, Japan. [4]Department of Surgery, Saitama Children's Medical Center, Saitama, Saitama, Japan. [5]Division of Surgery, Department of Surgical Specialties, National Center for Child Health and Development, Setagaya-ku, Tokyo, Japan. [6]Department of Pathology, National Center for Child Health and Development, Setagaya-ku, Tokyo, Japan. [7]Organ Transplantation Center, National Center for Child Health and Development, Setagaya-ku, Tokyo, Japan. [8]Department of Clinical Research, Saitama Children's Medical Center, Saitama, Saitama, Japan. [9]Department of Pediatric Surgery, Kyoto Prefectural University of Medicine, Kyoto Kamikyo-ku, Kyoto, Japan. [10]These authors contributed equally: Nanae Miyazaki, Shohei Takami. ✉e-mail: ykanai@g.ecc.u-tokyo.ac.jp

cystic ducts. The intrahepatic bile ducts are connected to the extrahepatic duct system of heterogenous characteristics and origins: right and left hepatic ducts, common hepatic duct, common bile duct, and gallbladder-cystic duct. In the developing extrahepatic bile duct system, SRY related HMG-box 17 (SOX17) plays a critical role in the development of the gall-bladder and cystic duct in the ventral foregut area[2,3]. Loss of SOX17 activity in the biliary bud results in the failure of gallbladder-cystic duct structure formation, observed in both *Sox17* conditional knockout mice and gallbladder-deficient rats[2–5]. The gallbladder wall, in particular, has a unique composition within the extrahepatic bile duct systems[6–8], relying on the SOX17-positive biliary epithelial stem/progenitor for the maintenance and regeneration in the gallbladder wall during perinatal stages. Recent advances in single-cell RNA sequencing highlighted the contrasting populations of SOX17-positive gallbladder wall progenitor and SOX17-negative (SOX9-positive) progenitor in organoids derived from different bile duct regions[9,10]. The proper formation of the gallbladder wall is dependent on the gene dose-dependent activity of *Sox17* during early-to-late stages of organ development[4,8]. Notably, in *Sox17*[+/−] embryos, severe autonomous atrophy of gallbladder walls, characterized by reduced cell proliferation activity and shedding of epithelial cells into the bile duct lumen, causes cholestasis and cholangitis in the liver and subsequent perinatal death in approximately 70% of *Sox17*[+/−] mouse neonates[8,11]. These findings suggest that autonomous defects in *Sox17*[+/−] gallbladder epithelia indirectly contribute to the liver injury during the perinatal period. However, the precise mechanisms underlying the progression to severe neonatal hepatic injury remain unclear.

In human, Biliary atresia (BA) is a devastating perinatal disease characterized by progressive obliterative cholangiopathy and scar formation with fibrosis of the extrahepatic bile duct (EHBD), accompanied by anomalous gallbladder shape and wall[12–16]. Despite being a complex and multifactorial disease, in 70–80% of cases no apparent defects found in other tissues or organs[16–18]. Theories abound with suggested causes ranging from toxins to viral infections to genetic susceptibility[15,19–21]. What is clear is that obstruction of EHBD can cause subsequent liver injury in infants, leading to liver fibrosis and lifelong health problems[13,22,23]. However, the exact mechanisms behind the onset and propagation of hepatic bile duct injury in BA patients during the fetal and perinatal period remain a mystery. In particular, we found that approximately 40% of BA cases are characterized by gallbladder wall hypoplasia with reduced SOX17 expression (i.e., SOX17-low cases)[24]. Furthermore, the ectopic presence of peribiliary glands, a feature typically associated with non-gallbladder bile duct walls and its progenitors[25], is commonly observed in BA gallbladders, mirroring the findings in the *Sox17*[+/−] mouse model[24]. These findings suggest possible defects in SOX17-positive gallbladder wall progenitors that are replaced by other bile duct progenitors in SOX17-low BA cases.

In this study, we investigate how hypoplastic gallbladder wall causes hepatobiliary pathogenesis in the *Sox17*[+/−] mouse embryos. Moreover, we conduct a retrospective study to analyze the impact of gallbladder wall hypoplasia on liver dysfunction and clinical outcomes in some BA patients at and after Kasai surgery. We show that the mechanism by which destruction spreads through the common hepatic duct to the intrahepatic bile ducts when abnormalities such as hypoplasia occur in the gallbladder wall due to reduced SOX17 function in the developing perinatal liver. The present findings indicate that adequate growth of the gallbladder wall during perinatal liver growth is essential for the formation of a normal network of extrahepatic and intrahepatic bile ducts in the developing liver.

## Methods
### Animal care and use
Animals were provided with water and commercial laboratory mouse chow ad libitum and were housed under controlled lighting conditions (daily light from 07:00 to 19:00). The pregnant mother mice and their fetuses was used (E13.5-E18.5). *Sox17*[+/−] mouse lines[26] were intercrossed and maintained at N10-11 backcross generations with C57B6 mice[4,8,11] (wild-type control: *Sox17*[+/+] littermates). *Shh*-cre; *Sox17*[flox/-] mice were also used by mating with *Shh*-cre[27], *Sox17*[flox 4], and *Sox17*[+/−] mouse (wild-type control: *Sox17*[flox/flox]

or *Sox17*[flox/+]). The survival rate of *Sox17*[flox/-] pups was approximately 31% at postnatal 3 weeks, similar to a *Sox17*[+/−] colony[11]. *Sox17*[+/−(GFP) 28] [wild-type control: *Sox17*[+/+] littermates], the *Alb*-cre[29], *ROSA*[mTmG 30] and ICR strain (Japan SLC, Japan) mice were also used in this study. We already confirmed the high *Shh*-cre activity in gallbladder-cystic duct progenitors using *Shh*-cre; *ROSA*[mTmG] and *Shh*-cre; *Sox17*[flox/flox] mice without any appreciable defects in other organs/tissues except for gallbladder and cystic duct. Each embryo was used as the experimental unit. We used as minimal animals as possible available for each experiment based our prior report[4]. Animals were monitored by veterinarian daily. Humane endpoint was determined based on a body condition score, posture, respiration, and activity. Investigators were blinded to group allocation during data collection and analysis.

All animal experiments were performed in strict accordance with the Guidelines for Animal Use and Experimentation of the University of Tokyo. All procedures were approved by the Institutional Animal Care and Use Committee of the Graduate School of Agricultural and Life Sciences at the University of Tokyo (approval ID P14-877, P18-121 and P20-035).

### Antegrade cholangiography to visualize bile flow by *in-utero* fetus injection
The intrauterine fetal injection is performed under isoflurane gas anesthesia and the pain killer[31]. In brief, the embryos were injected through the uterine wall into the fetal abdominal cavity with Fast green FCF (FG; blue color dye) or the mixture of FG and Cholyl-lysyl-fluorescein[32] (CLF; 451041; Corning, NY, U.S.A.) using the microcapillary glass needles (see Fig. 1a). In some pregnant mothers, intrauterine fetuses in one uterine horn were injected with ursodeoxycholic acid (UDCA; 100 µg/g embryonic BW) and FG to experimentally increase the fetal bile flow. Fetuses in another uterine horn were injected only with FG as control group. The abdominal wall of each operated pregnant mother was closed and allowed to recover for an appropriate time under monitoring by the veterinarian. The success of the injection was determined to be intraperitoneal based on the degree of the blue coloration observed through the uterus. The embryos injected in different locations were excluded.

### Retrograde Ink tracer experiment to visualize the intrahepatic bile duct network
Fetal livers from E18.5 were examined by carbon ink injection to visualize intrahepatic bile duct networks as reported[33]. The whole liver organ with the gallbladder/bile duct system and duodenum was isolated from wild-type or *Sox17*[+/−] embryos at E18.5. The Ink (SPC-200; Platinum Japan) was injected into duodenum using cannula at mild pressures. Thereafter the livers were fixed in 4% PFA-PBS for 12 h at 4 °C. Finally, the samples were cleared by the Cubic method[34] and observed under a stereomicroscope (SZX12, Olympus). We defined the injection as successful if it flowed into the intrahepatic bile ducts and excluded all others.

### Histology, lectin histochemistry and immunohistochemistry for a mouse model
For whole-mount immunohistochemistry staining, livers with extrahepatic ducts were fixed in 4% paraformaldehyde (PFA)-PBS for 12 h at 4 °C, then washed with PBS-Tween20. They were dehydrated and stored in 100% methanol at −20 °C. The samples were incubated with rhodamine-labeled dolichos biflorus agglutinin (DBA)-lectin (10 µg/ml; Vector Laboratories, RL-1032) and mouse anti-SMA antibodies (1/100 dilution; Sigma-Aldrich, A5228) for 12 h at 4 °C as previously described[4]. Signals were detected using secondary antibodies conjugated to Alexa-488 (1/100 dilution; Abcam, ab150113) for fluorescence microscopy (SZX16 and BX51N-34-FL2; Olympus).

The mouse PFA-fixed samples were embedded in paraffin or OCT compound and serially sectioned at a thickness of 4 or 7–20 µm, respectively. All sections were subjected to conventional histological (Masson's trichrome staining) and immunohistochemical staining. For immunohistochemical and lectin histochemical staining, sections were incubated with mouse anti-E-cadherin (E-cad) (1/250 dilution; BD, 610181), rabbit polyclonal anti-Green fluorescent Protein (GFP) (1/200 dilution, MBL, 598; 1/50 dilution,

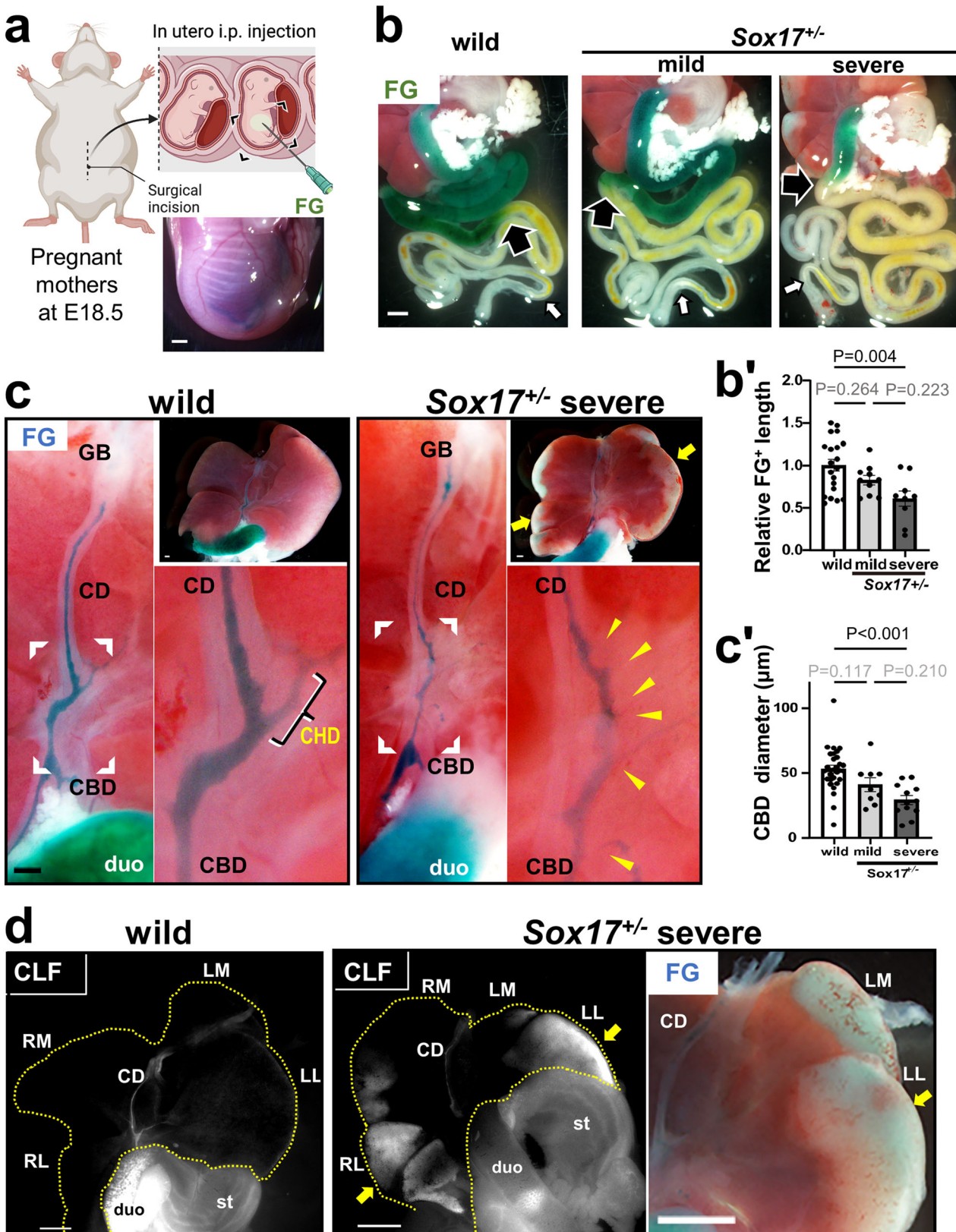

MBL, M048-3), and goat anti-SOX17 (1/100 dilution; R&D Systems, AF1924) antibodies and DBA lectin (5 μg/ml, Vector Laboratories, RL-1032). The reactions were visualized using a biotin-conjugated secondary antibody (1/400 dilution: Vector Laboratories, BA5000) and ABC Kit (Vector laboratories, PK-6100), or secondary antibodies conjugated with Alexa-488 (1:400 dilution, Invitrogen, A-21202, A-21441) or Alexa-594 (1:400 dilution, Invitrogen, A-11032) for fluorescence microscopy (BX51N-34-FL2, Olympus) or confocal laser microscopy (TCS SP8, Leica). DBA lectin-stained serial sections were then reconstructed into their 3D images by using ImageJ FIJI software v. 1.53t/ Java 1.8.0_172 (National Institutes of Health, MD)[35].

**Fig. 1 | Imaging of bile flow in intrauterine *Sox17*[+/−] embryos by a cholangiography. a** Schematic illustrations of Intrauterine embryos injected intraperitoneally with Fast green FCF (FG) to visualize the bile flow in vivo. The lower right inset shows the image of the isolated embryo with intraperitoneal FG injection. **b, c** FG outflows into the fetal intestine in the *Sox17*[+/−] embryos with (severe) or without (mild) liver damages, together with wild-type littermate, at embryonic days (E) 18.5 at 90 min after FG injection. In **b**, large black arrow and small white arrow indicate the reachable edge of FG (blue) and endogenous bile (yellow), respectively. In **c**, the FG-positive (FG+) extrahepatic biliary structure in the enlarged view of the hepatic hilar area represented by the bracket, showing the multiple small hepatic ducts (yellow arrowheads) in the severe *Sox17*[+/−] embryo, instead of the large common hepatic duct (CHD) in the wild-type littermate. Bar graph in (**b'**) shows relative FG+ gut length (wild-type: *n* = 20, mild *Sox17*[+/−]: *n* = 10, severe *Sox17*[+/−]: *n* = 9; maternal: *n* = 6). The bar graph in **c'** shows FG+ minimum luminal diameter of the common bile duct (CBD) (wild-type: *n* = 30, mild *Sox17*[+/−]: *n* = 9, severe *Sox17*[+/−]: *n* = 12; maternal: *n* = 8). Data are presented as mean ± s.e.m (by one-way ANOVA followed by Tukey's test). **d** Whole liver images of wild-type and severe *Sox17*[+/−] embryos injected at E18.5 with a mixture of FG and Cholyl-lysyl-fluorescein (CLF; fluorescein-labeled bile acid), showing ectopic accumulation of both two bile tracers in the peripheral hepatic lobules (yellow arrows). CBD common bile duct, CD cystic duct, CHD common hepatic duct, duo duodenum, GB gallbladder, LL left lateral lobe, LM left medial lobe, RM right medial lobe, RL right lateral lobe, st stomach. Scale bars, 1 mm (**a, b, d**), 250 μm (**c**).

## Liver enzyme assays

Mouse fetal serum was separated by centrifugation of whole blood at 12,000 rpm for 20 min. Serum alkaline phosphatase (ALP), alanine aminotransferase (ALT), aspartate aminotransferase (AST), and direct bilirubin (D-bil) levels were analyzed by SRL Diagnostics (Tokyo, Japan).

## Human BA samples and immunohistochemistry for SOX17/SOX9 index

We collected 115 gallbladder samples during Kasai portoenterostomy or primary liver transplantation (8 cases from 2018 to 2021 at the University Hospital of Kyoto Prefectural University of Medicine, 70 cases from 2000 to 2021 at the Saitama Children's Medical Center, 5 cases from 2020 to 2021 at the National Center for Child Health and Development, and 32 cases from 1995 to 2021 at the University of Tokyo Hospital). Among the 115 samples, the 61 new cases with well-preserved gallbladder walls were subjected to immunostaining using an anti-SOX17 (1/100; R&D Systems, AF1924) and rabbit anti-SOX9 (1/1000; Millipore, AB5535) antibody. The reactions were visualized using a biotin-conjugated secondary antibody (1/400 dilution: Vector Laboratories, BA5000 and BA1000) and ABC Kit (Vector laboratories, PK-6100), then used for estimation of the SOX17/SOX9 index in the gallbladder body, as previously described[24]. These gallbladders were analyzed together with the 13 gallbladders from a prior work[24] (Fig. 4a). In this study, we newly estimate the 'normal' gallbladder wall state was set as more than 70% SOX17/SOX9 index at the base on the mean value and range of the SOX17/SOX9 index of non-BA (110.3% [68.4–189.2%]) and control (136.4% [103.5–164.3%]) gallbladder walls[24]. Scientists were blinded to group allocation during analysis.

Clinical data of BA infants were obtained by retrospective review of medical records. The ethics committees of the participating institutes (Kyoto Prefectural University of Medicine, approval ID ERB-G-117, University of Tokyo, approval ID 2021060G; Saitama Prefectural Children's Medical Center, approval ID 2020-06-020; and National Center for Child Health and Development, approval ID 2021001) approved the study protocol. All participants in this study or their legal guardians signed consent forms or were given opportunity to opt-out of the study. We tried to contact all participants and get written informed consent. For the participants who were difficult to contact, we guaranteed the opportunity to refuse research cooperation by disclosing research information at each institute's website (The university of Tokyo: http://pedsurg.umin.jp/research, Kyoto Prefectural University of Medicine: http://pedsurg.kpu-m.ac.jp/news/index.php?seq=1, Saitama Prefectural Children's Medical Center: https://www.saitama-pho.jp/scm-c/rinsho/kenkyukadai_2020_06.html, National Center for Child Health and Development: https://www.ncchd.go.jp/center/information/epidemiology/index.html).

## Morphometry

Both FG-positive (FG+) gut length and total gut length were measured using ImageJ FIIJI software v. 1.53t/ Java 1.8.0_172 (National Institutes of Health, MD) at the base of gross anatomical images. Relative FG+ intestine length (FG+ length / total gut length) in *Sox17*[+/−] embryos relative to wild-type ones of littermate were also calculated. The viability of the fetus was assessed based on its heartbeat or movement, and deceased fetuses were excluded.

The FG+ diameter of the common bile ducts was measured using ImageJ FIIJI software v. 1.53t/ Java 1.8.0_172 (National Institutes of Health, MD) at the base of gross anatomical images. We excluded embryos in which FG could not be detected in the bile ducts because it had flowed entirely through the intestine.

As for the morphometry of the mouse gallbladder, the width (i.e., maximum diameter) of the presumptive gallbladder (i.e., a distal sack structure), the gallbladder-cystic duct length and the minimum diameter of the common bile duct, were measured using the liver stained with DBA wholemount immunohistochemistry (the below described) as previously described[11].

As for the phenotype of intrahepatic bile duct networks, the number of E-cad-positive cells around portal veins was calculated separately in the bile duct with or without apical lumen by using E-cad/DAPI-stained sections. The relative cell number with apical luminal surface per total E-cad-positive cells was estimated in wild-type and *Sox17*[+/−] embryos.

Liver degeneration area was semi-automatically measured using ImageJ software v. 2.3.0 (National Institutes of Health, MD) at the base of gross anatomical whole-liver images. In brief, gross anatomical whole-liver images were split into the color channels (red, green, and blue) [ImageJ command tab: Image-Color-Split channel]. Using the red channel images, the degeneration area at the liver edge was estimated as the upper 15% of brightness area [Image-Adjust-Threshold].

In human samples, gallbladder length, width, and gallbladder-cystic duct length were measured using gross pathological photographs taken during Kasai surgery. Additionally, the relative gallbladder width per gallbladder-cystic duct length or per gallbladder length were calculated.

## Statistics and reproducibility

Exact numbers of animal and human patients are provided in the Figure legend. Fisher's exact test, Tukey's honestly significant difference test, Tukey's multiple comparison test, Welch's *t* test, Student's *t* test, the Spearman rank correlation test, and the log-rank test were performed using Prism9 (GraphPad Software) and JMP Pro software v.16.0 (SAS Institute, NC). A *p* value ≤ 0.05 was considered indicative of statistical significance.

## Reporting summary

Further information on research design is available in the Nature Portfolio Reporting Summary linked to this article.

## Results

### Cholangiography in intrauterine *Sox17*[+/−] embryos

First, we have established a novel cholangiography method to the best of our knowledge by performing a single peritoneal injection into intrauterine mouse embryos with Fast green FCF (FG; a non-toxic visible contrast blue agent utilized worldwide as a color additive in food[36,37]. Moreover, in intravenous administration to adult rats, more than 90% of FG is excreted in bile[38], mimicking Cholyl-lysyl-fluorescein (CLF); Supplementary Fig. 1a–c) (Fig. 1a), which enables not only the visualization of the first bile outflow from the fetal liver at around E16.5 (Supplementary Fig. 1d), but also the estimation of bile flow levels as the reachable distance of FG+ signals along the fetal intestine (see Supplementary Fig. 2a–c).

**Fig. 2 | Ectopic formation of multiple hilar hepatic ducts and extrahepatic herniation of intrahepatic duct walls in severe *Sox17+/−* embryos prior to birth. a, b** Multiple small hepatic ducts (yellow arrowheads) visualized by retrograde cholangiography using black ink injection from the fetal duodenum (**a, a'**), anti-E-cadherin (E-cad) immunostaining (**b**; green fluorescence) and 3D construction of the DBA (Dolichos biflorus agglutinin; bile duct marker) lectin stained serial sections (**b'**) in *Sox17+/−* embryos at embryonic days (E) 18.5. In the wild-type embryo, one large common hepatic duct (CHD) connects to the cystic duct (red arrow). The bar graph in **a'** shows the numbers of hilar hepatic ducts per whole liver (mean ± s.e.m; by one-way ANOVA followed by Tukey's test [wild-type: *n* = 14, mild *Sox17+/−*: *n* = 4, severe *Sox17+/−*: *n* = 7, maternal: *n* = 7]). **c** Schematic representation of *Sox17+/+* and *Sox17+/−* mice (biliary atresia [BA] model) in an *Alb-*cre; *ROSA*^mTmG (Albumin [Alb] – Green Fluorescent Protein [GFP]+) background (left in **c**). Whole mount Alb- GFP+ liver (right in **c**) and immunostaining (**c'**) of intrahepatic (green; anti-GFP staining) and extrahepatic (magenta; DBA or anti-E-cad staining) bile duct walls at the hepatic hilus of E18.5 embryos. The ectopic Alb-GFP+ cells in hepatic ducts are protruding (white arrowhead) from the hepatic parenchyma (dashed lines). Note the contribution of several GFP+ cells in the damaged duct walls in *Sox17+/−* embryos (black arrows). The insets show the enlarged view of the area represented by the bracket. CD cystic duct, CBD common bile duct, CHD common hepatic duct, HD hepatic duct, LL left lateral lobe, LM left medial lobe, RM right medial lobe, PV portal vein, GB gallbladder, duo duodenum, Scale bars, 250 µm (**a**), 100 µm (**b**), 25 µm (**c**).

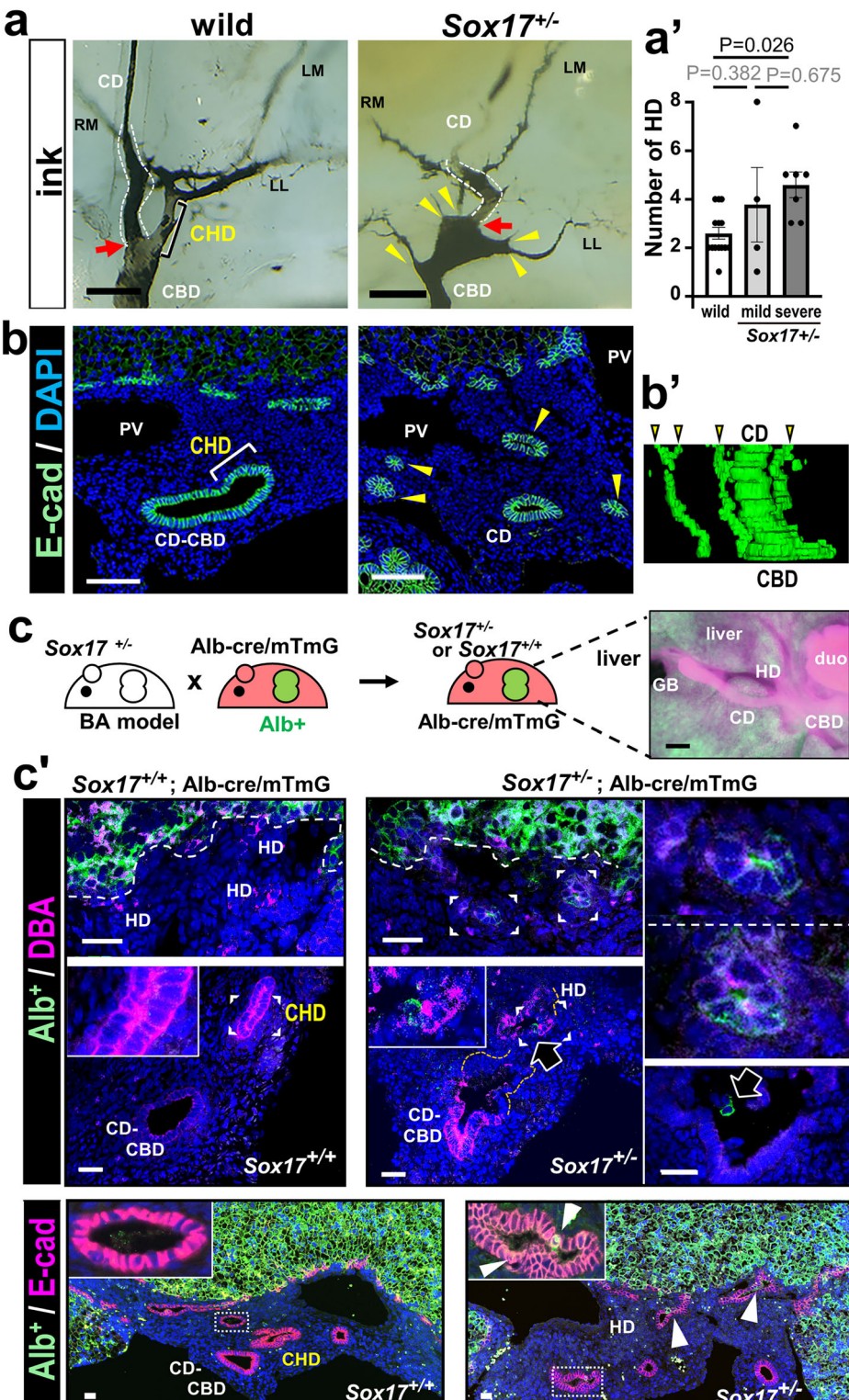

Approximately 70% of the *Sox17+/−* mouse die at neonatal stages with BA-like symptom, in contrast to remaining 30% showing a normal life span (over 1 year old) with normal fertility in males (N10-11 backcross to C57B6)[11]. Since the 65.4% of the *Sox17+/−* embryos used at E18.5 showed the gross-anatomical hepatic damages in this study, we separately examined the bile flow levels of these *Sox17+/−* embryos with ('severe' group) or without ('mild' group) the hepatic damages at E18.5[8]. As a result, under the antegrade cholangiography, the 'severe' *Sox17+/−* embryos showed a significant reduction of the bile output level (i.e., FG+ length in intestine), as compared with those in the wild-type ones just prior to birth (Fig. 1b, b'). Moreover, the 'severe' *Sox17+/−* embryos showed a significant narrowing of the common bile duct compared to the wild-type (Fig. 1c, c'). Most interestingly, instead of a large common hepatic duct in the wild-type livers, several small FG-positive hepatic ducts (yellow arrowheads in Fig. 1c) were ectopically observed draining at the hepatic hilus, while FG and CLF (fluorescein-labeled bile acid) were accumulated in the peripheral hepatic lobules

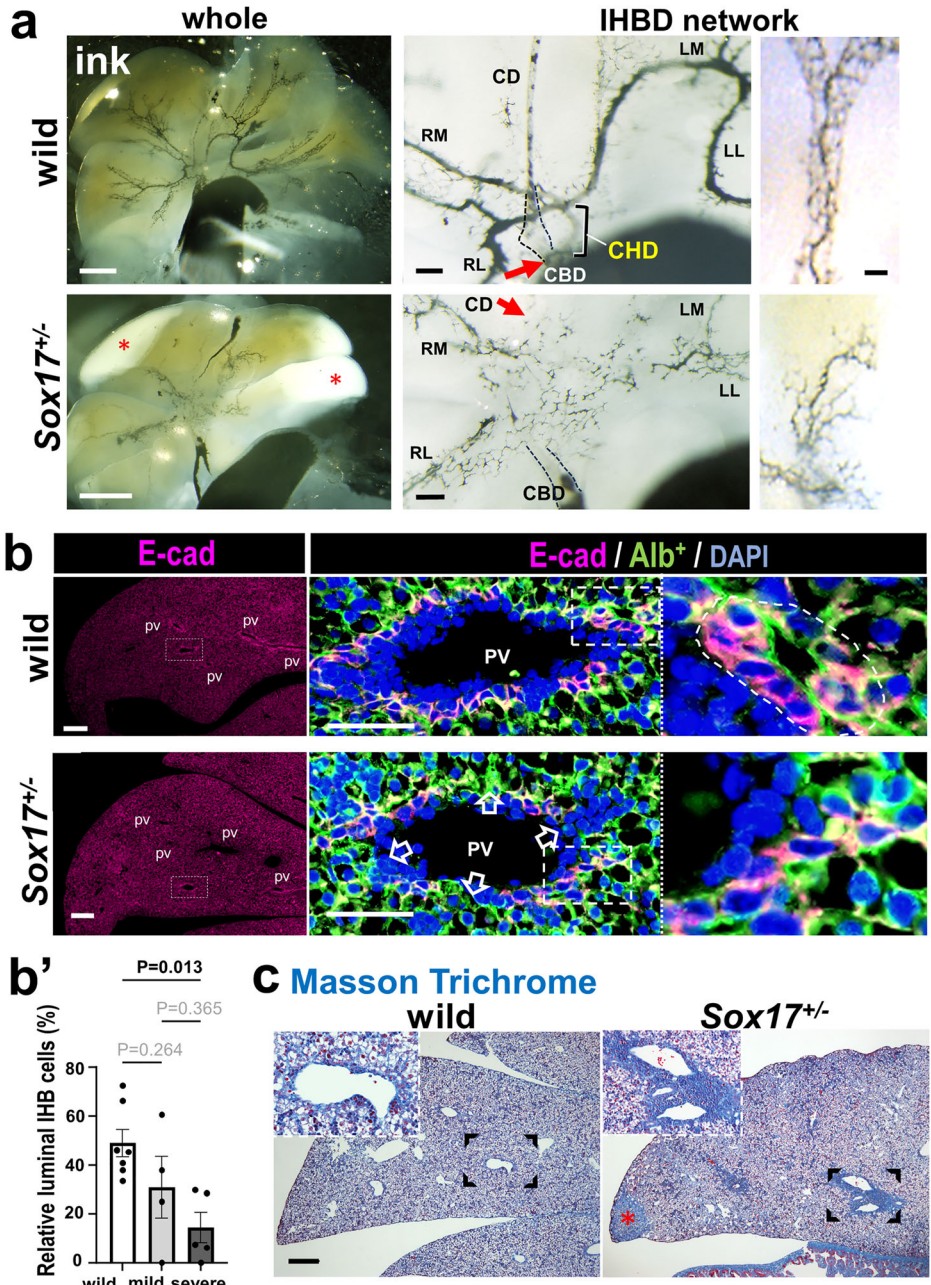

**Fig. 3 | Cloud-like fine networks of intrahepatic bile ducts, together with early signs of hepatic fibrosis in the severe $Sox17^{+/-}$ embryos prior to birth.**
**a** Retrograde cholangiography (black ink), showing a cloud-like intrahepatic bile duct (IHBD)network in the severe $Sox17^{+/-}$ embryo at embryonic days (E) 18.5. Note the poor development of networks near the damaged peripheral edge (red asterisk). Red arrows show that common hepatic duct (CHD) connects to the cystic duct (red arrow). **b**, **b'** Anti-E-cadherin immunostaining and the morphometric analyses of the stained sections, showing the reduced cell population of intrahepatic ducts with apical lumen per total E-cad+ cells in $Sox17^{+/-}$ embryos, as compared with wild-type littermates. In **b** Alb (Albumin)-GFP (Green fluorescent Protein)+ (green)/E-cad (E-cadherin)+ (red) intrahepatic ducts (yellow for double positive) around portal vein in the peripheral

lobular regions of the $Sox17^{+/+}$ and $Sox17^{+/-}$ embryos at E18.5 (*Alb*-cre; $ROSA^{mTmG}$ background). The construction of IHBDs in the $Sox17^{+/-}$ embryos is defective (open arrows). In **b'**, the bar graph shows the relative E-cad+ cell population of luminal duct per total cells (mean ± s.e.m.; by one-way ANOVA followed by Tukey's test [wild-type: $n = 7$, mild $Sox17^{+/-}$: $n = 4$, severe $Sox17^{+/-}$: $n = 5$; maternal: $n = 8$]). **c** Masson's tri-chrome in liver lobules of severe $Sox17^{+/-}$ embryos. In **c**, Masson trichrome staining sections, showing the fibrosis (blue) in the peripheral margins (red asterisk). CBD common bile duct, CD cystic duct, CHD common hepatic duct, IHBD intrahepatic bile duct, PV portal vein, RL right lateral lobe, RM right medial lobe, LM left medial lobe, LL left lateral lobe. Scale bars, 1 mm (left in **a**), 200 μm (middle in **a**, left in **b** and **c**), 100 μm (right in **a**), 50 μm (middle in **b**).

---

(Fig. 1d). This suggests aberrant hilar bile duct network and peripheral cholestasis in the severe $Sox17^{+/-}$ embryos prior to birth.

In addition, in the 'mild' $Sox17^{+/-}$ embryos without visible hepatic damages, both FG+ intestine length and minimum diameter of common bile duct exhibited intermediate values between wild-type and the 'severe' subgroup of $Sox17^{+/-}$ embryos, albeit of no significant difference between

the $Sox17^{+/-}$ mild group and the wild-type or $Sox17^{+/-}$ severe group (middle bars in Fig. 1b', c').

**Aberrant hilar and intrahepatic bile ducts**
Such aberrant multiple hilar hepatic ducts may be indirectly caused by gallbladder wall hypoplasia, since *Sox17* expression is restricted to the

gallbladder-cystic duct walls in the hepatobiliary development throughout the fetal stages[3,8] (Supplementary Fig. 3). First, hepatic duct structure at the hepatic hilus of the wild-type and *Sox17*[+/−] embryos at E18.5, was analyzed by retrograde ink tracer from the duodenum[33], in combination with the transparency CUBIC method[34]. Luminal black inks could visualize one major common hepatic duct ('CHD' in Fig. 2a) at the border between gallbladder-cystic duct and common bile duct at the hepatic hilus in the wild-type livers (red arrow in Fig. 2a), similar to those in the adult mouse[34]. This common hepatic duct branches the major bile ducts into the right and left lobules at the hepatic hilus (Fig. 2a). In the *Sox17*[+/−] embryos, multiple small hepatic ducts were ectopically formed at the liver hilus by the retrograde ink tracer (arrowheads in Fig. 2a) and the 3D reconstruction image using serial sections (Fig. 2b, b'). Together with the multiple small FG+ bile ducts at the hepatic hilus by the antegrade cholangiography (arrowheads in Fig. 1c), *Sox17*[+/−] gallbladder wall hypoplasia may affect the hepatic duct walls, leading their aberrant morphogenesis in the common hepatic duct via the hilar junction of the cystic duct.

Next, in order to examine the effects of gallbladder wall hypoplasia on the extra- and intrahepatic duct junction, we comparatively examined the distribution of Albumin (Alb)-GFP+ intrahepatic duct cells at the hepatic hilus between *Sox17*[+/−] and *Sox17*[+/+] embryos in an *Alb*-cre; *ROSA*[mTmG] mouse background[4] (Fig. 2c). The Alb-GFP+ intrahepatic duct walls, as well as Alb-GFP+ hepatocytes, are usually located within the liver parenchyma in the *Sox17*[+/+]; *Alb*-cre; *ROSA*[mTmG] embryos (broken line in Fig. 2c'), together with no contribution of Alb-GFP+ cells in the extrahepatic duct walls (left lower plate in Fig. 2c'). In contrast, in the *Sox17*[+/−] embryos, the Alb-GFP+ bile duct walls were ectopically protruded toward to extrahepatic duct from the hepatic parenchyma, suggesting extrahepatic herniation of the intrahepatic duct walls. In some cases, some Alb-GFP+ cells were found to contribute the walls of the large extrahepatic ducts (right two lower panels in Fig. 2c'). Since the hypoplastic gallbladder-cystic duct was continuously connected to the Alb+ intrahepatic duct via the aberrant hilar multi-hepatic ducts even in the 'severe' *Sox17*[+/−] embryos, it is therefore likely that the *Sox17*[+/−] gallbladder wall hypoplasia causes not only the aberrant formation of the small multi-hepatic ducts, but also extrahepatic herniation of the intrahepatic duct walls at the hepatic hilar region. This in turn suggests that the intrahepatic duct walls may compensate for the damaged hilar duct walls caused by the gallbladder wall hypoplasia in the *Sox17*[+/−] embryos.

In the affected liver lobules, cloud-like immature intrahepatic duct networks were evident in the proximal sites of the hepatic lobules in the 'severe' *Sox17*[+/−] embryos (Fig. 3a). Morphometric analysis of E-cad staining sections revealed a significant reduced population in the ductular structure with open lumen per total E-cad-positive cells (Fig. 3b, b', Supplementary Fig. 4), suggesting the immature intrahepatic duct structure without apical lumen in the 'severe' *Sox17*[+/−] embryos as compared with wild-type ones. In some severe cases, Masson's trichrome showed considerable positive signals, albeit of weak signals, around the intrahepatic ducts (Fig. 3c), suggesting the early onset of the liver fibrosis prior to birth in these *Sox17*[+/−] embryos. These findings suggest that gallbladder hypoplasia may affect the intrahepatic duct maturation via the hilar intra- and extrahepatic duct connections.

## Effects of GB hypoplasia in human BA
In the previous pilot study, around 40% BA patients showed reduction in SOX17-positive gallbladder epithelia, which were replaced by SOX9-positive epithelia at the time of Kasai surgery[24], indicating the defective gallbladder wall progenitors in these cases similar to those in the *Sox17*[+/−] mouse model. Next, to retrospectively analyze the hepatic degeneration in human BA with SOX17-low gallbladder wall hypoplasia, we used morphometric parameters of human gallbladder samples that were excised at the Kasai surgery, instead of less-visible hepatic hilar tissues at the time of diagnosis made (Fig. 4a), and then examine the relationship between their morphometric parameters and clinical liver cholestatic markers. In this study, we newly obtained 61 BA gallbladder samples with well-preserved

gallbladder epithelia (total 74 samples, including the image/morphometric data [i.e., gallbladder width, and length of gallbladder-cystic duct at the time of Kasai surgery]; Fig. 4a'), examined the relative SOX17 level (SOX17/SOX9 index) by immunohistochemistry (Fig. 4a"), and then selected 24 BA gallbladder samples with aberrantly reduced SOX17 expression (SOX17-low; <70% SOX17/SOX9 ratio; Fig. 4a).

In the 24 BA gallbladders with SOX17-low expression, we examined the correlations between the morphometric data of gallbladder-cystic duct and hepatic serum markers at the time of Kasai surgery (Fig. 4b, c; Supplementary Fig. 5a–d). As a result, in the SOX17 low group, thicker gallbladder width was clearly correlated with lower serum D-bil, AST and ALT levels in the SOX17-low group (Fig. 4b; Supplementary Fig. 5a), in addition to the high correlation between serum D-bil level and relative gallbladder width per either gallbladder-cystic duct or gallbladder length (Fig. 4c; Supplementary Fig. 5b, c). This is in contrast to no correlation between gallbladder width and these hepatic damage markers in the other 50 BA cases with a normal level of SOX17 expression (SOX17/SOX9 ratio, 118.94% [79.5–264.1%]) (Fig. 4b; Supplementary Fig. 5a–c). Moreover, gallbladder length showed no correlation with the serum markers in not only SOX17-low, but also in other cases (Supplementary Fig. 5d). Therefore, the reduced gallbladder width may directly contribute to the severity of liver injury at the time of Kasai surgery in the SOX17-low BA cases.

The gallbladder parameters, clinical outcomes, and patient characteristics of the aberrant SOX17-low group (*n* = 24) and others (*n* = 50) are provided in Supplementary Table 1 and Supplementary Fig. 6. Although most of the characteristics (e.g., age at Kasai surgery and BA type) showed no significant differences between the SOX17-low group and other BAs, the SOX17-low group clearly showed a better prognosis after Kasai surgery, i.e., higher native liver survival (Supplementary Fig. 6a) and lower incidence of liver transplantation (Supplementary Table 1). In addition, there were also no significant differences between BA types[39] and gallbladder parameters and no significant correlation between age at Kasai surgery and SOX17/SOX9 (Supplementary Fig. 6b, c). These results suggested that the better prognosis observed in the SOX17-low group might not be attributed to other prognostic factors previously reported, such as BA types and age at Kasai surgery[40,41].

## Effects of excessive GB wall hypoplasia in the BA model mice
As mentioned above, the reduced gallbladder width was closely correlated with severe liver damage in human SOX17-low BA cases. To examine the correlation of the gallbladder width with gallbladder wall hypoplasia, we used the *Sox17*[flox/−] mouse in combination with a gallbladder-cystic duct wall specific deletion by *Shh*-cre line (Fig. 5a, b; Supplementary Fig. 7). In these *Shh*-cre; *Sox17*[flox/−] (flox/−) embryos at E18.5, *Shh*-cre-dependent loss of additional *Sox17* flox allele causes different levels of hypoplastic gallbladder wall (upper panels in Fig. 5a), together with the reduced SOX17 expression (lower panels in Fig. 5a). By morphometrical analyses, *Shh*-cre; flox/− embryos, albeit of considerable variation among the identical genotypes, showed a significant reduction in gallbladder width, but not alter the gallbladder-cystic duct length, as compared with wild-type and flox/− littermates (Fig. 5b). These data confirm that the gallbladder width is a suitable morphometric parameter of SOX17-low gallbladder wall hypoplasia even in the mouse *Sox17*-mutant model. In contrast, the measurement of gallbladder length was not effective in assessing gallbladder wall hypoplasia. This is because the boundary between the gallbladder and the cystic duct region could not be accurately determined, even with the use of anti-smooth muscle actin (SMA) staining (green fluorescence in upper panels of Fig. 5a), suggesting a lack of correspondence between the gallbladder wall agenesis and the length of gallbladder/gallbladder-cystic duct.

Most interestingly, in this gallbladder wall agenesis model, the reduced gallbladder width, a morphometric parameter of gallbladder wall hypoplasia, was closely associated with severe liver injury (i.e., serum ALP level and liver degeneration area; Fig. 5c). This is in contrast to no association between the minimum dimeter of the common bile duct and liver damage.

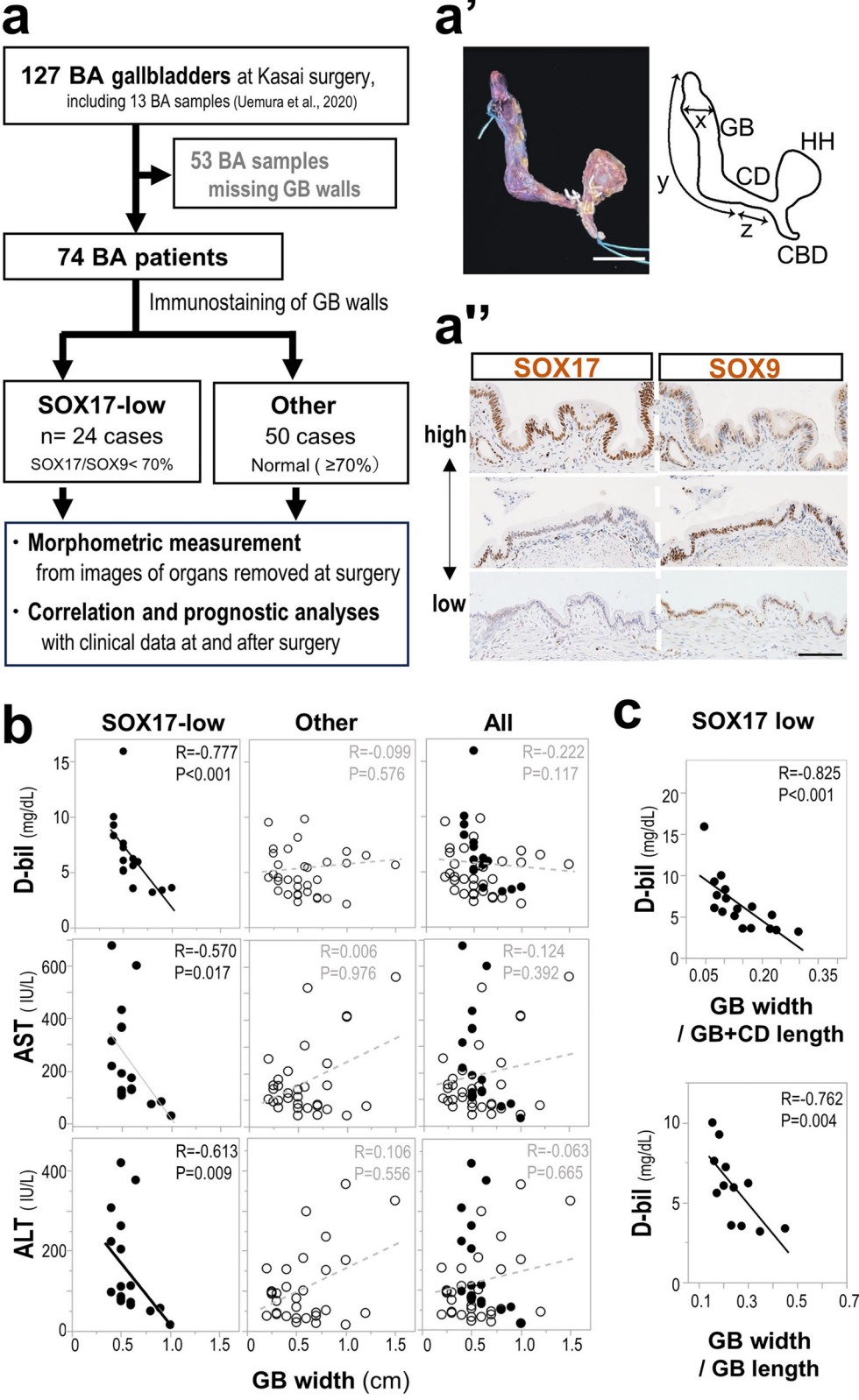

Based on these findings, we concluded that the gallbladder wall hypoplasia directly correlates with liver damage levels in human SOX17-low BA cases and its *Sox17*$^{+/-}$ mouse model. This is consistent with our observations of a better prognosis following the removal of the hypoplastic gallbladder during the Kasai surgery in the SOX17-low group (Supplementary Fig. 6; Supplementary Table 1).

## Discussion

This study highlights a novel insight, to the best of our knowledge, into liver damage via the 'hilar' hepatic duct, which is primarily caused by defects in the SOX17-positive gallbladder wall in the perinatal stage (Fig. 5d). In summary, gallbladder wall atrophy affects the proper morphogenesis of the large common hepatic duct, leading to the ectopic formation of multiple

**Fig. 4 | Impact of gallbladder hypoplasia on the liver injury in human BA cases with aberrant SOX17-low gallbladder walls at the time of Kasai surgery. a-a",** Study design to analyze the characteristics and prognosis of the Biliary atresia (BA) cases with aberrant SOX17-low gallbladder walls. Of 127 gallbladder specimens from BA patients, 74 showed preserved gallbladder walls at the time of Kasai surgery. Among the 74 samples, 24 gallbladders were classified as having aberrant gallbladder walls with reduced SOX17 expression level (less than 70% of the SOX17/SOX9 index) by Immunohistochemical analysis of SOX17 and SOX9 (brown color). In **a'**, gross anatomical image and scheme of an extrahepatic duct resected by Kasai surgery in a BA patient. CBD: common bile duct, CD: cystic duct, GB: gallbladder, HH: hepatic hilum, x = GB width, y = GB length, z = CD length. In **a"**, human BA

gallbladder walls of two serial sections of gallbladder body that were stained with anti-SOX17 and anti-SOX9 antibody (SOX17/SOX9 indices = 115.9%, 56.3% and 6.5% from top [high] to bottom [low]). **b, c,** Spearman correlation coefficients between gallbladder width (**b**, GB width [cm]; **c,** relative GB width per gallbladder-cystic duct [GB + CD] length or per GB length) and serum markers (direct bilirubin [D-bil, mg/dl], aspartate aminotransferase [AST, IU/l] or alanine aminotransferase [ALT, IU/l]) in the SOX17-low BA cases (solid circle), other cases (open circle), and both of them (All) (SOX17-low: $n = 17$, Other: $n = 34$, All: $n = 51$). Each point shows the value (black solid line, $p < 0.01$; gray solid line, $0.01 \leq p < 0.05$; gray dashed line, $p \geq 0.05$). Scale bar, 1 cm (**a'**), 100 μm (**a"**).

**Fig. 5 | Effects of excessive gallbladder wall hypoplasia on gallbladder shape and liver injury in *Sox17*-mutant embryos just prior to birth.**
**a** Whole-mount dolichos biflorus agglutinin (DBA; magenta) and anti-alpha-smooth muscle actin (SMA; green)-double staining (upper panels) and anti-SOX17 immunostaining (brown) of sagittal sections of distal sac-like structures (i.e., presumptive gallbladder; lower panels) of *Sox17*^flox/flox^ (f/f), *Sox17*^flox/−^ (f/−) and *Shh*-cre; *Sox17*^flox/−^ (*Shh*-cre; f/−) embryos at embryonic days (E) 18.5. The border between gallbladder and cystic duct walls in the *Shh*-cre; f/− embryos cannot be defined even by anti-SMA staining for gallbladder-specific smooth muscle layers in the presumptive gallbladder region. asterisks, vascular smooth muscle.
**b** Dot plots of the gallbladder (GB) width (i.e., maximum diameter of DBA-positive distal sac-like structure), gallbladder-cystic duct (GB + CD) length and minimum common bile duct (CBD) diameter (y-axis) in three genotypes (i.e., control [f/+ and f/f], heterozygous [f/− and +/−] and homozygous [*Shh*-cre; f/−] deletion of two *Sox17* alleles (x-axis). Note significant reduction in GB width and CBD diameter by Tukey's honestly significant difference test), in contrast to no change in GB + CD length among three genotypes (*Shh*-cre; f/−: $n = 9$, f/− or +/−: $n = 14$, f/+ or f/f: $n = 9$).
**c** Spearman rank correlation tests between GB width, relative GB width per GB + CD length and minimum CBD diameter in x axis and liver injury level (i.e., serum Alkaline phosphatase [ALP, IU/l] level and degeneration area) on the y-axis (black solid line, $p < 0.01$; gray solid line, $0.01 \leq p < 0.05$; gray broken line, $p \geq 0.05$) ($n = 13$). **d** Schematic illustration of the ripple effects of GB wall hypoplasia on the intrahepatic duct (IHBD) network. Hypoplastic GB wall with reduced SOX17 expression causes reduced GB width, albeit of no change in GB + CD length (left in **d**), and it simultaneously causes abnormal formation of a hilar bile duct network (right in **d**) as follows: i) deformation of a large common hepatic duct (i.e.; multiple small hepatic ducts); ii) extrahepatic herniation of the IHBD wall; iii) a cloud-like immature IHBD network near the hepatic hilus; and iv) peripheral cholestasis. CBD common bile duct, CD cystic duct, GB gallbladder, IHBD intrahepatic bile duct, HD hepatic duct. Scale bar in **a**, 100 μm.

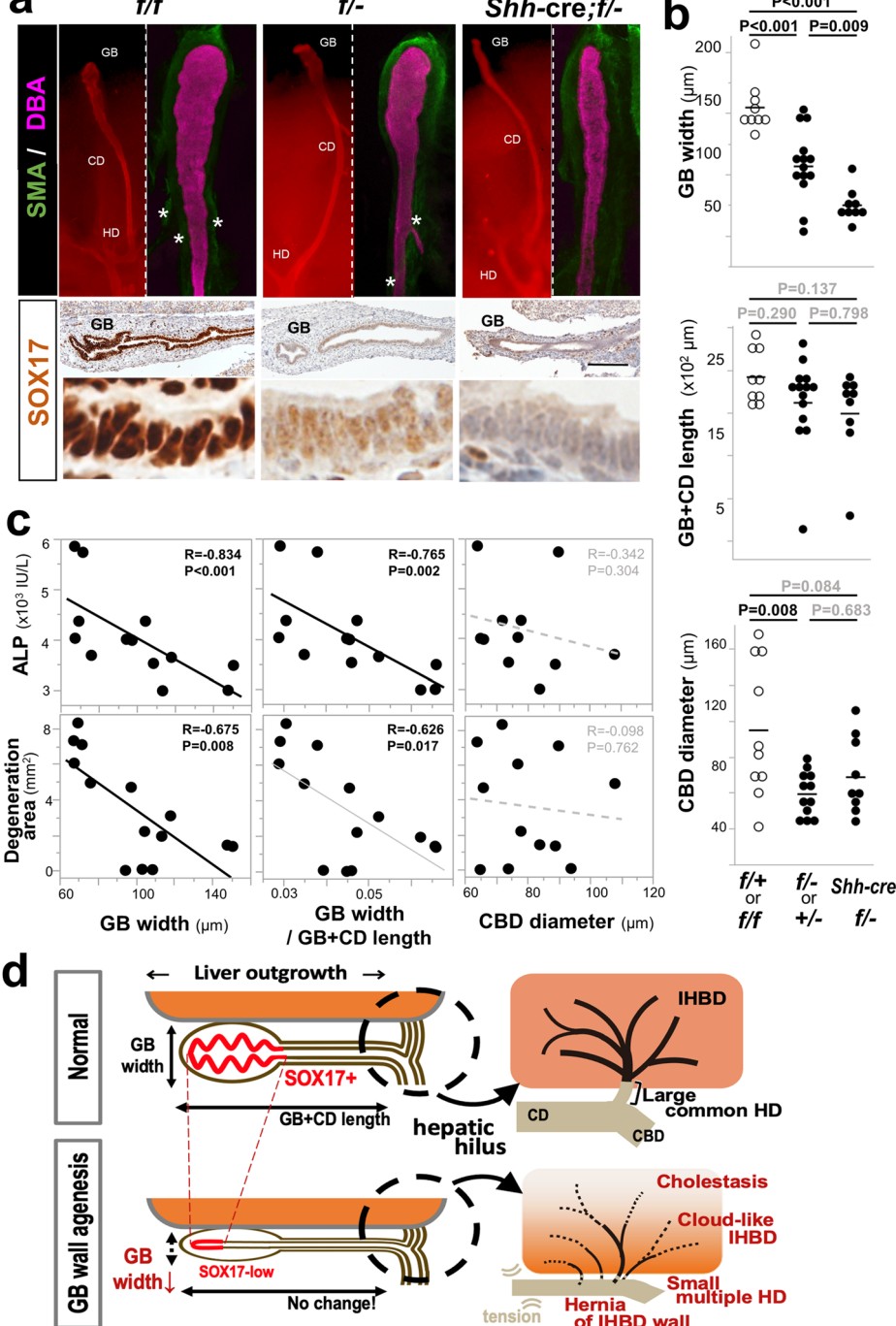

small hepatic ducts at the hepatic hilus. This may be due to the compensation of the hilar hepatic duct wall for hypoplasia of the reduced gallbladder-cystic duct wall. Such deformed hilar hepatic ducts cause aberrant extrahepatic herniation and cloud-like bile duct network of intrahepatic bile duct walls, leading to severe cholestasis prior to birth.

In SOX17-low cases (approximately 40% of human BA cases), the reduced gallbladder width, a morphometric parameter of gallbladder wall hypoplasia, was closely associated with severe cholestasis and liver damage at the Kasai surgery. Such a close correlation was demonstrated in *Sox17* mutant BA mice. In this study, gallbladder hypoplasia resulting from decreased expression of SOX17 causes malformation in *Sox17* haploinsufficiency mice, including the 'cloud-like' bile duct network and the thin branched hepatic ducts at the hepatic hilus, similar to the finding of some human BAs[42–44]. Taken together with the ectopic PBG (niche of stem cell in hepatic ducts[25,45,46]) appearance in both human BA and mouse model[24], these findings clearly imply that *Sox17* heterozygous mouse model mimics the findings of human BA cases. In SOX17-low BA, gallbladder wall hypoplasia might possibly contribute to the progressive obliterative cholangiopathy by a similar mechanism in the *Sox17*-mutant mouse model; however, it is impossible to prove what actually happened in the early stages that led to the scarring of the porta hepatis.

It is speculated that BAs are initiated by various factors such as virus infection[47], toxins[48] and genetic defects[8], suggesting the presence of the various subtypes even in the isolated BAs. A recent genome-wide association study (GWAS) of 811 human BA genome samples also supports the involvement of the multiple genes including *SOX17*, to contribute to the onset and/or exacerbation of BA[49]. In this study, SOX17-low group showed a better prognosis after Kasai surgery, i.e., higher native liver survival rate and lower incidence of liver transplantation, together with no significant correlation of the SOX17-low group with the known prognostic factors such as BA types and patient's age at Kasai surgery. Therefore, together with the GWAS data showing the significant association of *SOX17* genes in human BA[49], it is also possible that SOX17-low BA may be a potential subtype of human BA. The reason for the better prognosis after the Kasai surgery in this group may be explained by resecting the gallbladder-hepatic duct as a trigger of the BA and prevent the progression of liver damage after the surgery. Alternatively, it is also possible that, in these cases, the healthy (SOX17-negative) bile duct walls in the hilar extrahepatic and/or intrahepatic ducts can regenerate and establish a better functionality when the hilum is connected to the small bowel to drain bile.

One of the most reliable features of BA is the presence of gallbladder abnormalities, such as an echogenic non-identical, atrophic, noncontractile and/or irregularly shaped gallbladder without a definable luminal wall[50–54] like the *Sox17* haploinsufficiency mice, together with a hilar hyperechoic zone and the triangular cord sign[50–52,54]. In contrast, approximately 10% of patients with BA have a normal gallbladder[55]. A recent study demonstrates the association between intrahepatic SOX9 expression (another SOX factor for cholangiocytes except for the gallbladder walls) and the prognosis in BA[56]. Considering that SOX17-negative (i.e., SOX9+ and/or PDX1+) bile duct progenitors are present in the extrahepatic bile ducts[45,46,57], these findings suggest a possible disruption of hepatic and common bile duct walls by the aberrant SOX9+ and/or PDX1+ progenitors in the pathogenesis of other BA groups[10].

Some animals, like the rats, deer, horses, and the pigeons, lack a sac-like gallbladder-cystic duct system[5,58,59], while it is a major component of the extrahepatic biliary system in the mouse and human fetus. BA is rarely found in animals without a gallbladder, such as pregnant mares and foals, which usually receive good veterinary care[60,61]. This highlights the role of this organ in potential causes of the disease, including toxins, viruses, and genetic mutations. Animals with a gallbladder - such as sheep[62], cows[62], dogs[63,64], monkeys[65], and cats[66] have all been shown to develop symptoms similar to those seen in human BA, supporting the present hypothesis that the presence of a gallbladder may contribute to the condition's onset. When examining the causes of biliary atresia in humans, it may be possible in the near future to consider the health of gallbladder wall progenitors as one of multiple contributing factors.

## Data availability

The authors confirm that all data underlying the findings are fully available without restriction. All relevant data are within the paper and its Supporting Information files. Source data for the figures are available as Supplementary Data 1.

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

## Acknowledgements

The authors thank Drs. K. Deie, T. Tajiri, E. Watanabe, Y. Sengoku, Y. Fukumura for their support to human samples. The authors also thank Drs. Y.

Hirate, N. Ota, W.Promust, S. Zhen, K. Hayakawa and T. Niimi for their technical advice and support. This study was supported by JSPS KAKENHI (24H00537, 24228005 and 20H00445) to Y.K., JSPS KAKENHI (18K14583) to M.U., JSPS Research Fellowship to N.M. (DC2 22J11160), JST SPRING, Grant Number JPMJSP2108 to N.M., JSS Young Researcher Award 2023, from Japanese Surgical Society to S.T., the University of Tokyo Excellent Young Researcher system to A.Y. and the Translational Research program; Strategic PRomotion for practical application of Innovative medical Technology, TR-SPRINT, from Japan Agency for Medical Research and Development, AMED Grant number: JP21lm0203003j0005 to J.F.

## Author contributions

N.M., S.T. and Y.K. designed the study. N.M., S.T., and M.U. performed the biological experiments. M.K.-A. supported for the mouse models. S.T., H.O., M.T., H.K., Y.K., T.Y., M.K., A.N., M.H. and J.F. provide BA samples and their clinical records. N.M. and S.T. analyzed the data. Every illustration in the figures was created by N.M., S.T., M.U., A.Y., and Y.K. The manuscript was written by N.M., S.T., M.U., H.O., A.Y., R.H., J.F. and Y.K. contributed with scientific discussions and interpretation of all experiments. The illustrations in Fig. 1a, Supplementary Figs. S1a and S2a were created with BioRender.com by A.Y. The illustrations in Figs. 2c and 5d were created with PowerPoint for Microsoft 365 by N.M. and Y.K.

## Competing interests

The authors declare no competing interests.
