## [Peer Review File · Communications Medicine]

Reviewers' comments:

Reviewer #1 (Remarks to the Author):

The authors analyzed the correlation between SOX17 dysfunction in a murine model and in human biliary atresia patients hypothesizing a correlation gallbladder dysplasia and SOX17 activity with liver damage / deterioration and biliary atresia outcome.

SOX17 is a well-known regulator for the differentiation of the biliary phenotype and its impact on cholangiopathies and biliary malignancies has been discussed. The association of gall bladder wall abnormalities in human BA and SOX \pm mice has already been discussed by the working group from Tokyo, concluding that there is evidence associating SOX17 reduction and the early pathogenesis of BA gallbladders.

While some novel analysis of the SOX17 \pm mice have been added compared to previous works of the working group, the translational aspect investigating the human specimens seems inconclusive. Clinical data is missing - are there differences in the age groups, could we conclude different SOX17 / 9 ratios in the older cohorts, focusing on the gallbladder with and types, there should be considerations of the Ohi BA types. Some distinct BA forms do not inherent a vanishing gallbladder, but seem to show a mucous (non-biliary) dilatation of the gallbladder not correlating with liver damage and deterioration. Therefore, the translation of those results on the human BA pathogenesis and the conclusion seems far-fetched.

Reviewer #2 (Remarks to the Author):

Brief summary of the manuscript:

The authors write about the effect of SOX-17 gene on the development of the intra- and extra hepatic biliary tree and gallbladder, and its possible association with biliary atresia.

Overall impression of the work:

Excellent, interesting, and well-presented work.

Specific comments, with recommendations for addressing each comment:

- The references are not combined at the end of the manuscript, instead they are presented after each section.
- I strongly recommend writing a very short summary about the embryology of the intra- and extra hepatic biliary tree in humans.
- Please write full names of ALT, AST, etc, in the text (then use abbreviations).
- Page 3 line 54: “then passes through the interconnected ducts of increasing” I recommend using: then passes through the canaliculi to the interconnected ducts of increasing.
- Page 3 line 70: “approximately 70% of Sox17+/- neonates”. You mean the mice neonates? (please make it clear).
- Page 3 line 74-75: “characterized by blockage and inflammation of the extrahepatic bile duct (EHBD) at the hepatic hilus” BA is best described as the obliteration and scar formation with fibrosis of EHBD, but there is no inflammation. Not all BA cases are at the hepatic hilus.
- Page 3 line 76: “complex and multifaceted disease”. Do you mean multifactorial?
- Page 3 line 79: “inflammation in EHBD can cause devastating liver injury”. The obstruction/obliteration of the EBHD (not inflammation) causes bile accumulation in liver, and bile is toxic and induces inflammation.
- Page 4 line 100: “Approximately 70% of the Sox17+/- mouse are lethal at neonatal stages”. Do you mean they suffer from the lethal condition (the mouse itself is not lethal).
- Page 4 line 102: “embryos used at E18.5”. I recommend explaining what is E.18.5.

- Page 4 line 105: “albeit not ‘mild’ ones”. What do you mean by this sentence?
- Page 4 line 116: “albeit of no significant difference among them”. Among whom (wild, severe, and mild)?
- Page 5 line 123 you used “wildtype” and in page 6 line 155 you used “wild-type”. I recommend unifying.
- Page 6 line 161: “around 40% BA patients showed gallbladder wall hypoplasia with a considerable reduction in SOX17-positive gallbladder epithelia”. Do you mean 40% of BA patients showed gallbladder wall hypoplasia, OR 40% of BA patients WITH gallbladder wall hypoplasia showed considerable reduction in SOX17-positive gallbladder epithelia?
- Page 6 line 166: you used “KASAI”. I recommend Kasai.
- Page 6 line 170: what do you mean by “length of length”?
- Page 6 line 176-177: “gallbladder width was clearly correlated with serum D-bil, AST and ALT levels in the SOX17-low group”. Would you please make it clearer that the width is “inversely” correlated with liver damage in SOX-17 low group (thicker width means less liver injury).
- Page 7 line 189-190: “the SOX17-low group clearly showed a better prognosis after Kasai surgery”. Very interesting! What do you think is the reason?
- Page 7 line 210-211: “the reduced gallbladder width, a morphometric parameter of gallbladder wall hypoplasia, were closely associated with severe liver”. I recommended using was closely (not were closely).
- Page 7 line 213 “the common bile duct and liver damages”. I recommended using damage (not damages).

- Page 8 line 216: “following damaged gallbladder removal during the Kasai surgery”. I suggest using: following the removal of the hypoplastic gallbladder during the Kasai surgery.
- Page 8 line 217-218: “supports the notion that gallbladder wall hypoplasia may be a contributing factor in human cases of biliary atresia”. Please provide a reference.
- Page 8 line 230: “the ‘cloud-like’ bile duct network at the hepatic hilus as an atypical symptom”. I recommend using the word finding (not symptom).
- Page 8 line 236: ‘Some animals, like the rat, the deer, the horse, and the pigeon’. I suggest using “Some animals, like rats, deer, horses, and pigeons”.
- Page 26 line 478: “mice were also used by the mating with Shh-cre38”. I recommend removing “the”.
- Page 27 line 499: “Fetus livers”. I suggest using : Fetal livers.
- Page 28 line 538-539: “were subjected to immunostaining using an anti-SOX17 (1/100; R&D Systems, AF1924) or rabbit anti-SOX9 (1/1000; Millipore, AB5535) antibody. Do you mean “and” (I would assume that all samples were stained for both SOX17 and SOX9).

We thank all reviewers for their helpful comments on our manuscript. We believe their guidance has allowed us to make a much stronger manuscript. Please see individual responses below in blue. The sentences of the revised manuscript are in red.

Regarding the PBD (patent blue dye) we used in this study, its chemical name 'Fast green FCF (FG)' will be disclosed from 1st April 2024. We, therefore, used FG instead of PBD in this revised manuscript. We have added the following explanation to: "Fast green FCF(FG)..... more than 90% of FG is excreted in bile," (p8 line 218-p.9 line 220).

The editor's suggestions:

'We would ask that you address the reviewers' concerns in full. In particular, a revised manuscript should include a broader discussion of how your manuscript provides novel insight to the field and perhaps toning down of some claims regarding relevance to the human condition if the mouse data is not completely supportive.'

We have added three new paragraphs on a novel insight and discussion to the BA (p.13 line 357-p.14 line390, in the revised manuscript), together with the clinical data on age and BA type, as shown in the new Supplementary Figure 6 and Supplementary Table 1 in the revised manuscript. Additionally, we have toned down some clauses in the human BA in response to reviewer comments.

Reviewer #1 (Remarks to the Author):

The authors analyzed the correlation between SOX17 dysfunction in a murine model and in human biliary atresia patients hypothesizing a correlation gallbladder dysplasia and SOX17 acitivity with liver damage / deterioration and biliary atresia outcome.

SOX17 is a well-known regulator for the differentiation of the biliary phenotype and its impact on cholangiopathies and biliary malignancies has been discussed. The association of gall bladder wall abnormalities in human BA and SOX+/- mice has already been discussed by the working group from Tokyo, concluding that there is evidence associating SOX17 reduction and the early pathogenesis of BA gallbladders.

We are grateful to reviewer #1 for offering valuable insights and constructive suggestions that have greatly improved the quality of our paper. We appreciated the overview of the BA gallbladders in the previous reports including our research data. As evidenced in the subsequent responses, we have incorporated all these comments and recommendations, including clinical data, into the revised manuscript.

While some novel analysis of the SOX17+/- mice have been added compared to previous workes of the working group, the translational aspect investigating the human specimens seems inconclusive.

Clinical data is missing –

1) are there differences in the age groups, could we conclude different SOX17 / 9 ratios in the older cohorts,

There are no differences in the age at Kasai surgery between SOX17-low and other group: 61.3 ± 26.3 (SOX17-low) vs 59.8 ± 32.3 (Other). We have added this data as new Supplementary Table 1. Following this comment, we have newly also added the Spearman correlation coefficients data showing no correlation ($p= 0.578$; $r=-0.066$)

between by the SOX17/SOX9 and age at the time of Kasai surgery (p.11. line 317-p.12 line 321 in the revised manuscript; new Supplementary Figure 6b).

- 2) focusing on the gallbladder with and types, there should be considerations of the Ohi BA types.

We have newly added the Ohi BA types in new Supplementary Table 1 and showed no significant difference between SOX17-low group and the other group (p=0.945). We also compared the gallbladder parameters (length, width and relative gallbladder width /gallbladder-cystic duct) between BA types. There were no differences between any combinations. (p.11. line 317-p.12 line 321 in the revised manuscript; new Supplementary Figure 6c).

- 3) Some distinct BA forms do not inherent a vanishing gallbladder, but seem to show a mucous (non-biliary) dilatation of the gallbladder not correlating with liver damage and deterioration. Therefore, the translation of those results on the human BA pathogenesis and the conclusion seems far-fetched.

We have added three paragraphs to the discussion (p.13 line 357-p.14 line 390). We clarified that our claim is restricted to some part of human BA with reduced SOX17 expression at gallbladder epithelia (p.13 lines 369-380) and discussed different causes and potential multiple subtypes of human BA, including do not inherent a vanishing gallbladder (p.13, line 381-p.14 line 390). We also added the explanation of a recent GWAS data using 811 BA cases, which includes the SOX17 gene as a candidate modifier gene in J Hepatology, 79(6), December 2023, 1385-1395 (p.13, lines 370-373). Finally, to respond to this comment, we toned down of the last sentence of the discussion as follows: **'When examining the causes of biliary atresia in humans, it may be possible in the near future to consider the health of gallbladder wall progenitors as one of multiple contributing factors.'** (p.14, lines 398-400)

Reviewer #2 (Remarks to the Author):

The authors write about the effect of SOX-17 gene on the development of the intra- and extra hepatic biliary tree and gallbladder, and its possible association with biliary atresia.

Excellent, interesting, and well-presented work.

We are grateful to reviewer #2 for the critical comments and useful suggestions that have helped us to improve our paper. As indicated in the responses that follow, we have taken all these comments and suggestions into account in the revised version of our paper.

We appreciate the overview of the effect of SOX17 on the development of bile ducts and its relationship with biliary atresia.

- 1) The references are not combined at the end of the manuscript, instead they are presented after each section.

We combined the references at the end and correct the reference numbers.

- 2) I strongly recommend writing a very short summary about the embryology of the intra- and extra hepatic biliary tree in humans.

we have added short summary about the embryology of the intra- and extrahepatic biliary tree in humans and mice (p.4, lines 59-63)

- 3) Please write full names of ALT, AST, etc, in the text (then use abbreviations).
We have corrected to the full name in the text then used abbreviations as follows: **SRY related HMG-box 17 (SOX17)** (p.4 line 65), **dolichos biflorus agglutinin (DBA)** (p.6, line 138), **alkaline phosphatase (ALP)** (p.7 line 158), **alanine aminotransferase (ALT)** (p.7, line 158), **aspartate aminotransferase (AST)** (p.7, lines 158-159), **direct bilirubin (D-bil)** (p.7, line 159), **Albumin (Alb)** (p.10 line 260, p.21 line 635, p.22 line 650), **E-cadherin [E-Cad]** (p.6 lines 145-146, p.22 line 650), **Green Fluorescent Protein [GFP]** (p.6 line 146, p.21 line 635, p.22 line 650) and **FG-positive (FG+)** (p.7 line 183, p.21 line 613) throughout the revised manuscript and figure legend.
- 4) Page 3 line 54: “then passes through the interconnected ducts of increasing” I recommend using: then passes through the canaliculi to the interconnected ducts of increasing
We have corrected in the revised manuscript as follows, “**then passes through the canaliculi to the interconnected ducts of increasing**” (p.4 line 58 in the revised manuscript).
- 5) Page 3 line 70: “approximately 70% of Sox17^{+/-} neonates”. You mean the mice neonates? (please make it clear).
We mean that the phrase “approximately 70% of Sox17^{+/-} neonates” refers to “mouse neonates”. We have changed the sentence to “**approximately 70% of Sox17^{+/-} mouse neonates**” (P.4 line 78 in the revised manuscript) for the clarification.
- 6) Page 3 line 74-75: “characterized by blockage and inflammation of the extrahepatic bile duct (EHBD) at the hepatic hilus” BA is best described as the obliteration and scar formation with fibrosis of EHBD, but there is no inflammation. Not all BA cases are at the hepatic hilus.
We have corrected the sentence to “**In human, biliary atresia (BA) is a devastating perinatal disease characterized by progressive obliterative cholangiopathy and scar formation with fibrosis of the extrahepatic bile duct (EHBD), accompanied by anomalous gallbladder shape and wall.**” (p.4 lines 82-84 in the revised manuscript).
- 7) Page 3 line 76: “complex and multifaceted disease”. Do you mean multifactorial?
We have corrected the sentence to “**complex and multifactorial disease**” (p.4 line 85 in the revised manuscript).
- 8) Page 3 line 79: “inflammation in EHBD can cause devastating liver injury”. The obstruction/obliteration of the EBHD (not inflammation) causes bile accumulation in liver, and bile is toxic and induces inflammation.
We have corrected the sentence to “**What is clear is that the obstruction of EHBD can cause cholestasis and subsequent liver injury in infants, leading to liver fibrosis and lifelong health problems.**” (p.4 lines 87-88 in the revised manuscript).
- 9) Page 4 line 100: “Approximately 70% of the Sox17^{+/-} mouse are lethal at neonatal stages”. Do you mean they suffer from the lethal condition (the mouse itself is not lethal).
We mean that the phrase refers to the fact that 70% of Sox17^{+/-} mice suffer from a severe liver condition before birth and die immediately after birth. We have rephrased this sentence as follow: “**Approximately 70% of Sox17^{+/-} mice die at neonatal stages with BA-like symptoms.**” (p.9 line 224 in the revised manuscript).

- 10) Page 4 line 102: “embryos used at E18.5”. I recommend explaining what is E.18.5.
We mean that “E18.5” is an abbreviation for “embryonic day 18.5”. We have corrected the first sentence (“embryonic day 8.5 [E8.5]”) in the revised manuscript. (p.4 lines 60-61 in the revised manuscript)
- 11) Page 4 line 105: “albeit not ‘mild’ ones”. What do you mean by this sentence?
We mean that the phrase “albeit not ‘mild’ ones” refers to the observation that mice in the *Sox17*^{+/-} mild group did not exhibit a significant reduction in bile output levels compared to the wild-type. We have deleted this sentence in the revised manuscript.
- 12) Page 4 line 116: “albeit of no significant difference among them”. Among whom (wild, severe, and mild)?
We mean that the phrase refers to showing no significant difference between the *Sox17*^{+/-} mild group and the wild-type or *Sox17*^{+/-} severe group. We modified this sentence into “albeit of no significant difference between the *Sox17*^{+/-} mild group and the wild-type or *Sox17*^{+/-} severe group” in the revised manuscript (p.9 lines 241-242 in the revised manuscript).
- 13) Page 5 line 123 you used “wildtype” and in page 6 line 155 you used “wild-type”. I recommend unifying.
We have unified the notation for the ‘wild-type’ in the revised manuscript.
- 14) Page 6 line 161: “around 40% BA patients showed gallbladder wall hypoplasia with a considerable reduction in SOX17-positive gallbladder epithelia”. Do you mean 40% of BA patients showed gallbladder wall hypoplasia, OR 40% of BA patients WITH gallbladder wall hypoplasia showed considerable reduction in SOX17-positive gallbladder epithelia?
We have deleted “gallbladder wall hypoplasia with a considerable”.
- 15) Page 6 line 166: you used “KASAI”. I recommend Kasai.
We have changed “KASAI” to “Kasai” in the revised script.
- 16) Page 6 line 170: what do you mean by “length of length”?
We have deleted “the length of”.
- 17) Page 6 line 176-177: “gallbladder width was clearly correlated with serum D-bil, AST and ALT levels in the SOX17-low group”. Would you please make it clearer that the width is “inversely” correlated with liver damage in SOX-17 low group (thicker width means less liver injury).
We have corrected the sentence to “In the SOX17 low group, thicker gallbladder width was clearly correlated with lower serum D-bil, AST, and ALT levels.” (p.11 lines 301-302 in the revised manuscript).
- 18) Page 7 line 189-190: “the SOX17-low group clearly showed a better prognosis after Kasai surgery”. Very interesting! What do you think is the reason?
Thank you for your interest in our results. We have classified BA to SOX17-low and the other BA. We think SOX17-low BA has a similar pathophysiology of the *Sox17* mutant BA mouse model. Because the SOX17 expression is restricted at EHBD, the center of the lesion in SOX17-low BA may be in the EHBD. Therefore, Kasai surgery is highly effective in SOX17-low BA. Some of the other BA may affect bile duct at liver hilus or intrahepatic bile duct. These lesions are difficult to remove by Kasai surgery.

Consequently, Kasai surgery may be less effective in the other BA. We inserted a new paragraph in the discussion (p.13 lines 357-p.14 line 390 in the revised manuscript) to express this speculation and our concern that there may be multiple causes for the observed differences in the other BA.

- 19) Page 7 line 210-211: “the reduced gallbladder width, a morphometric parameter of gallbladder wall hypoplasia, were closely associated with severe liver”. I recommended using was closely (not were closely).

We have corrected the sentence in the revised manuscript as follows, “the reduced gallbladder width, a morphometric parameter of gallbladder wall hypoplasia, was closely associated with severe liver” (p.12 line 341 in the revised manuscript)

- 20) Page 7 line 213 “the common bile duct and liver damages”. I recommended using damage (not damages).

We have corrected the sentence in the revised manuscript as follows, “the common bile duct and liver damage”. (p.11 line 343 in the revised manuscript)

- 21) Page 8 line 216: “following damaged gallbladder removal during the Kasai surgery”. I suggest using: following the removal of the hypoplastic gallbladder during the Kasai surgery.

We have corrected the sentence in the revised manuscript as follows, “following the removal of the hypoplastic gallbladder during the Kasai surgery” (p.12 line 346 in the revised manuscript).

- 22) Page 8 line 217-218: “supports the notion that gallbladder wall hypoplasia may be a contributing factor in human cases of biliary atresia”. Please provide a reference.

We incorporated our speculation and the summary of this study in this sentence., Therefore, we do not have any reference to this comment. We have deleted the latter part of this sentence to clarify that this statement is based on our speculation.

- 23) Page 8 line 230: “the ‘cloud-like’ bile duct network at the hepatic hilus as an atypical symptom”. I recommend using the word finding (not symptom).

We have corrected the sentence in the revised manuscript “the ‘cloud-like’ bile duct finding of some human BAs” (p.13 line 361-362 in the revised manuscript).

- 24) Page 8 line 236: ‘Some animals, like the rat, the deer, the horse, and the pigeon’. I suggest using “Some animals, like rats, deer, horses, and pigeons”.

We have corrected the sentence in the revised manuscript as follows, “Some animals, like rats, deer, horses, and pigeons”(p.14 line 391 in the revised manuscript).

- 25) Page 26 line 478: “mice were also used by the mating with Shh-cre38”. I recommend removing “the”.

We have corrected the sentence in the revised manuscript.

- 26) Page 27 line 499: “Fetus livers”. I suggest using : Fetal livers.

We have corrected the sentence in the revised manuscript as follows, “Fetal livers” (p.6 line 127 in the revised manuscript).

- 27) Page 28 line 538-539: “were subjected to immunostaining using an anti-SOX17 (1/100; R&D Systems, AF1924) or rabbit anti-SOX9 (1/1000; Millipore, AB5535) antibody. Do you mean “and” (I would assume that all samples were stained for both SOX17 and SOX9).

We have corrected the sentence in the revised manuscript as follows, “**were subjected to immunostaining using an anti-SOX17 (1/100; R&D Systems, AF1924) and rabbit anti-SOX9 (1/1000; Millipore, AB5535) antibody**” (p.7 lines167-168 in the revised manuscript).

Others

- 1) Reference numbers have been updated to accommodate the additional references.
- 2) We have corrected our manuscript according to the guidelines as follows:
 - A) We have corrected the order of affiliations (p.1 lines 3-6)
 - B) We have added a 'Plain Language Summary' (p.2 lines 44-53)
 - C) Provided the following sections: 'Introduction' 'Method' 'Results' and 'Discussion'. In the main.
 - D) We have corrected to the “Supplementary Fig” from” Extended data Fig”.
 - E) We have corrected the misprints.

REVIEWERS' COMMENTS:

Reviewer #1 (Remarks to the Author):

The authors have made substantial changes and have implemented all comments by the reviewers. Especially the discussion has benefited from the amendments and the additional analysis.

Therefore, there are no further concerns regarding publication in the journal.

Reviewer #2 (Remarks to the Author):

Brief summary of the manuscript:

The authors write about the effect of SOX-17 gene on the development of the intra- and extra hepatic biliary tree and gallbladder, and its possible association with biliary atresia.

Overall impression of the work:

Great work with substantial revision. I recommend publishing.

Specific comments, with recommendations for addressing each comment:

- Page 13 line 387 (discussion): “The reason for the better prognosis after the Kasai surgery in this group may be explained by resecting the gallbladder-hepatic duct as a trigger of the BA and prevent the progression of liver damage after the surgery.”

o I think it is a very interesting observation that SOX17-low BA have better Kasai results. Yet, I am not sure that we can explain why, maybe it is because at cellular level they are able to “regenerate” or establish a better functionality when the hilum is connected to the small bowel to drain bile? You don’t need to remove this sentence; I think we cannot know why, and it is better to provide multiple theories (which will open the horizon for future research in this interesting subgroup of BA).

- Page 14 line 397 (discussion): “supporting the notion that the presence of a gallbladder may contribute to the condition's onset.”

o Can you please provide a reference to the notion.